# Reverse2Complete: Unpaired Multimodal Point Cloud Completion via Guided Diffusion

Wenxiao Zhang
University of Science and Technology of China
Hefei, China
wenxxiao.zhang@gmail.com

Hossein Rahmani
Lancaster University
Lancaster, United Kingdom
h.rahmani@lancaster.ac.uk

Xun Yang
MoE Key Laboratory of Brain-inspired Intelligent
Perception and Cognition, University of Science and
Technology of China
Hefei, China
xyang21@ustc.edu.cn

Jun Liu*
Lancaster University
Lancaster, United Kingdom
j.liu81@lancaster.ac.uk

## ABSTRACT

Unpaired point cloud completion involves filling in missing parts of a point cloud without requiring partial-complete correspondence. Meanwhile, since point cloud completion is an ill-posed problem, there are multiple ways to generate the missing parts. Existing unpaired completion methods usually leverage generative adversarial training by transforming partial shape encoding into a complete one in the low-dimensional latent feature space. However, "mode collapse" often occurs, where only a subset of the shapes is represented in the low-dimensional space, reducing the diversity of the generated shapes. In this paper, we propose a novel unpaired multimodal shape completion approach that directly operates on point coordinate space. We achieve unpaired completion via a single diffusion model trained on complete data by "hijacking" the generative process. We further augment the diffusion model by introducing two guidance mechanisms to facilitate mapping the partial point cloud to the complete one while preserving its original structure. We conduct extensive evaluations of our approach, which show that our method generates shapes that are more diverse and better preserve the original structures compared to alternative methods.

## CCS CONCEPTS

• **Computing methodologies** → **Point-based models**.

## KEYWORDS

Point Cloud, Shpae Completion, Point Cloud Diffusion Model

*Corresponding author.

**ACM Reference Format:**
Wenxiao Zhang, Hossein Rahmani, Xun Yang, and Jun Liu. 2024. Reverse2Complete: Unpaired Multimodal Point Cloud Completion via Guided Diffusion. In *Proceedings of the 32nd ACM International Conference on Multimedia (MM '24), October 28-November 1, 2024, Melbourne, VIC, AustraliaProceedings of the 32nd ACM International Conference on Multimedia (MM'24), October 28-November 1, 2024, Melbourne, Australia*. ACM, New York, NY, USA, 10 pages. https://doi.org/10.1145/3664647.3680590

## 1 INTRODUCTION

The increasing accessibility of affordable sensors, such as LIDAR and depth cameras, has led to a surge of interest in 3D data within both the vision and robotics communities. Nevertheless, such scanned data cannot always be directly applied in real-world scenarios due to incompleteness caused by limited resolution and viewpoint occlusion in 3D scans[33, 34]. Hence, it is crucial to recover complete 3D shapes from partial ones which are immensely valuable for various vision-related applications[4, 5, 18, 30, 80, 81, 83].

Pioneered by PCN [76], learning-based point cloud completion methods [20, 23, 24, 31, 43, 45, 51, 52, 56, 57, 59, 63, 65, 66, 68, 75, 84, 87] have achieved impressive completion results. However, they depend on datasets containing both partial and corresponding complete shapes, which are challenging to obtain. To overcome this challenge, unsupervised point cloud completion has been proposed [2, 9, 58, 77]. In the unsupervised scenario, only unpaired samples from the partial point clouds and the complete shapes are available. Meanwhile, shape completion is an ill-posed problem because there can be multiple ways to generate missing parts for a given partial shape, particularly when the input is excessively incomplete. To address this issue, MPC [60] was first developed to handle unpaired multimodal shape completion, aiming to produce various complete shapes for a single partial shape to enhance output diversity. However, this is hard to achieve because the training data only includes one true complete shape for each partial shape. In MPC [60], a conditional GAN-based model was proposed to learn a one-to-many mapping from partial shapes to complete shapes. Following MPC [60], ShapeInversion[2] produces multiple completion results via GAN inversion by adjusting the sampled latent vector.

While MPC [60] and ShapeInversion [2] have made progress in unpaired multimodal shape completion, the diversity of generated

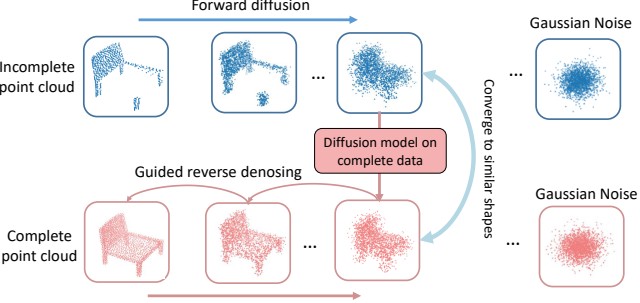

**Figure 1: The incomplete point cloud and its corresponding complete point cloud will gradually become more similar in shape as noise is added during the forward diffusion process. Based on this observation, we propose to "hijack" the forward diffusion process of the partial point cloud through a diffusion model trained on complete data to "reverse" to the complete shape by guided denoising process.**

shapes remains limited. Both approaches utilize GANs to model multimodality in high-dimensional shape space and map it to a low-dimensional latent space. However, a common issue that arises is "mode collapse", where only a subset of modes is represented in the low-dimensional space. Specifically, given a partial point cloud, the generated shapes tend to share similar structures, limiting the diversity of the results.

To address the issue of mode collapse, we propose a novel unpaired multimodal completion method that employs an unconditional diffusion model to perform point cloud completion in the point coordinate space, instead of mapping the partial data to complete ones in the latent space. The key idea behind our method is illustrated in Fig. 1. Our observation is that an incomplete and its corresponding complete point clouds will gradually become more similar in shape as noise is added during the forward diffusion process. Therefore, we can "hijack" the forward diffusion process of the partial point cloud through a diffusion model trained on complete data to generate the complete shape by the reverse denoising process. Specifically, in the forward diffusion process, we add an appropriate amount of noise to the partial point cloud to smooth out high-frequency signals while retaining the overall shape. We then gradually remove the noise via a single unconditional diffusion model trained on complete data during the reverse diffusion process. Finally, we obtain a denoised output that is similar to the partial input, but follows the distribution of the complete data.

During the reverse denoising process, we introduce *two guidance mechanisms* to facilitate the transformation from incomplete to complete point clouds. Firstly, we introduce a *structure preservation guidance* that promotes the denoised point cloud to retain most of the original shape of the partial input. Secondly, we propose a *classifier guidance* that encourages the denoised point cloud to conform more closely to the complete point cloud distribution.

Different from previous methods, MPC[60] and ShapeInversion [2], which achieve multimodal completion by sampling different latent vectors in the latent space as guidance, our approach incorporates diverse information directly into the partial input shape.

Specifically, we sample a reference shape from the pretrained diffusion model, and introduce a combination strategy to mix the partial input with the reference shape in the point coordinate. This mixed point cloud is then used as the diffusion input, which not only maintains the original shape information but also incorporates referenced shape structure.

Though recent studies on point cloud completion like [88] and [37] have used conditional diffusion with paired supervision to perform point cloud completion in a straightforward manner, which regards the partial input as the condition and the complete point as the target, it is still non-trivial and challenging to perform the diffusion processes for point cloud completion without paired supervision. To the best of our knowledge, our method is the first diffusion-based method for the task of completing point clouds without paired data. Comprehensive evaluations of our approach demonstrate that our method generates more diverse shapes than alternative methods while still preserving the input shape.

In summary, our contributions are as follows:

- We propose a novel unpaired multimodal point cloud completion method, which performs incomplete-to-complete mapping in coordinate space via an unconditional diffusion model, and further introduce a novel multimodal completion strategy, allowing our method to be guided by specified reference shapes.
- We propose two guidance mechanisms including classifier guidance and structure preservation guidance, to effectively direct the diffusion process towards producing more complete and faithful results.
- Our experimental results demonstrate that our method achieves state-of-the-art performance on unpaired multimodal completion with both synthetic and real-world datasets, and is capable of generating structurally diverse results while preserving the original shape.

## 2 RELATED WORK

**Supervised point cloud completion.** In the early years, researchers developed several effective descriptors, such as [25, 41, 50], which leverage geometric cues to fill in missing parts on a surface. Point cloud completion can also be achieved by utilizing a symmetry prior [40, 44, 53]. In addition, researchers have proposed data-driven methods, such as [26, 28, 47], which involve retrieving the most similar model based on partial input from a large 3D shape database.

In recent years, learning-based methods have often utilized a deep neural network with an encoder-decoder architecture to directly map partial input to a complete shape. Some pioneering works [14, 19, 27, 62, 70] rely on volumetric representations, allowing for direct application of convolution operations. In contrast, PCN [76] directly generates complete shapes from partial point clouds by decoding the global latent features. More recent works[3, 8, 20, 23, 24, 31, 38, 42, 45, 46, 51, 52, 54, 56, 63, 64, 66–68, 78, 79, 82, 84, 86, 88, 90] have focused on preserving observed geometric details from local features in incomplete inputs. Snowflakenet [65] introduces a snowflake point deconvolution for point cloud completion. More recently, there have been transformer-based completion methods. PoinTR [75] uses a geometry transformer to predict missing shapes, while SeedFormer [87] introduces

a new shape representation named Patch Seeds for shape completion. FBNet [69] refines the output by rerouting high-level information from the coarse output via a graph-based network. Proxyformer proposed a proxy alignment assisting strategy for point completion. Anchorformer utilized the anchor nodes for generating more discriminative results. SVDFormer[89] propose a self-view fusion network to enhance the completion results. These works perform completion in a supervised manner using both partial point clouds and their corresponding complete shapes. FSC [61] proposes a new setting to conduct completion when points are extremely sparse.

**Unsupervised point cloud completion.** While supervised methods have produced impressive completion results, they typically require large-scale datasets that include both incomplete and complete point clouds, which can be difficult to collect. As a pioneering work for unsupervised point cloud completion, Pcl2Pcl [9] proposes an adversarial learning-based approach to transform latent code of the incomplete shape into that of the complete shape. Cycle4Completion [58] introduces two cycle transformations for dual-direction completion. Himanshu et al. [1] proposes a multimodal point cloud completion method via conditional Implicit Maximum Likelihood Estimation (IMLE). Cai et al. [2] encode a series of related partial point clouds into a unified latent space that represents a complete shape code and multiple occlusion codes. Cui et al. [12] proposes an energy-based latent transport module aiming to model the distribution gap between the partial and the complete shape codes. MPC [60] handles unpaired multimodal shape completion via a variational autoencoder combined with GAN. Inspired by GAN inversion, ShapeInversion [77] searches for a latent code in the latent space of a pre-trained GAN to perform multimodal completion. KTNet[6] tries to solve this task from the new perspective of knowledge transfer. All these unpaired completion methods perform the incomplete-to-complete transformation in latent space.

**Diffusion Models for Point Cloud.** Diffusion models have emerged as an effective method for learning a data distribution[29, 71–74] that can be easily sampled from. Sohl-Dickstein et al. [48] introduced the diffusion model for generating images, and since then, several works [21, 49] have simplified and accelerated the approach. Diffusion models have also been applied to various tasks, such as image synthesis [16, 39, 85], 3D Gaussian[15, 35]point cloud generation [36, 88] and point cloud completion [10, 37, 88].

In the domain of point cloud completion, prior diffusion-based works [37, 88] have typically used conditional diffusion models under paired supervision to achieve point cloud completion in a straightforward way, where the incomplete point cloud serves as the input condition and the complete point cloud as the target. However, how to employ the diffusion model for unparied point cloud completion is non-trivial and remains underexplored. In this paper, we propose a novel approach that performs unpaired multimodal point completion via unconditional diffusion with delicately designed gradient guidances.

## 3 PRELIMINARY

Given a clean point cloud sampled from the real point cloud distribution $\mathbf{x}_0 \sim q(\mathbf{x}_0)$, a fixed Markov chain is established by following the forward process of the diffusion model. This process gradually introduces Gaussian noise to the initial point cloud $\mathbf{x}_0$ through a series of $T$ time steps, yielding a sequence of noisy point clouds $\mathbf{x}_1, \mathbf{x}_2, \cdots, \mathbf{x}_T$. Noise is added to the coordinates of points within point clouds, similar to how noise is added to individual pixels in images. The forward process can be mathematically denoted as:

$$q(\mathbf{x}_{1:T} \mid \mathbf{x}_0) := \prod_{t=1}^{T} q(\mathbf{x}_t \mid \mathbf{x}_{t-1}), \tag{1}$$

$$q(\mathbf{x}_t \mid \mathbf{x}_{t-1}) := \mathcal{N}\left(\mathbf{x}_t; \sqrt{1 - \beta_t}\mathbf{x}_{t-1}, \beta_t \mathbf{I}\right), \tag{2}$$

where the sequence $\beta_1, \ldots, \beta_T$ is a fixed variance schedule to control the noise's step sizes. In contrast, the reverse process constructs a Markov chain with Gaussian transitions, whose parameters are parameterized to iteratively eliminate the noise from the initial point cloud $\mathbf{x}_T$, which is sampled from a Gaussian distribution with zero mean and an identity covariance matrix. This process takes place over $T$ time steps and produces a clean point cloud:

$$p(\mathbf{x}_{0:T}) := p(\mathbf{x}_T) \prod_{t=1}^{T} p(\mathbf{x}_{t-1} \mid \mathbf{x}_t), \tag{3}$$

$$p_\theta(\mathbf{x}_{t-1} \mid \mathbf{x}_t) := \mathcal{N}\left(\mathbf{x}_{t-1}; \mu_\theta(\mathbf{x}_t, t), \sigma_t^2(\mathbf{x}_t, t)\mathbf{I}\right). \tag{4}$$

The denoising diffusion probabilistic models (DDPM) [21] utilize time-dependent constants by setting $\sigma_t(\mathbf{x}_t, t) = \sigma_t \mathbf{I}$. The parameterization of $\mu_\theta$ consists of a linear combination of $\mathbf{x}_t$ and $\epsilon_\theta(\mathbf{x}_t, t)$, where $\epsilon_\theta(\mathbf{x}_t, t)$ predicts the noise component in the noisy sample $\mathbf{x}_t$. By optimizing the variational bound of negative log-likelihood $\mathbb{E}\left[-\log p_\theta(\mathbf{x}_0)\right]$, the parameters of $\mu_\theta(\mathbf{x}_t, t)$ are learned. Following DDPM, the training objective $\mathcal{L}_{\text{simple}}$ reduces to a mean-squared error loss between the predicted and actual noise $\epsilon \sim \mathcal{N}(0, \mathbf{I})$ in $\mathbf{x}_t$:

$$\mathcal{L}_{\text{simple}} := \|\epsilon_\theta(\mathbf{x}_t, t) - \epsilon\|^2. \tag{5}$$

By deriving the training objective from the variational bound on the negative log-likelihood $\mathbb{E}\left[-\log p_\theta(\mathbf{x}_0)\right]$ of the data, the diffusion model is able to generate data that follows the source data distribution via a denoising process.

## 4 METHOD

The pipeline of our proposed model is illustrated in Figure 2. During the training stage, we train a single unconditional diffusion model $D$ on a set of complete point clouds using the same training setting as in PVD [88]. Additionally, we train a time-dependent binary classifier $C(\mathbf{p}, t)$ to guide the diffusion process.

During the testing stage, our method follows these steps: First, we propose an input combination strategy that incorporates the multimodal information into the partial input, enabling the generation of different results guided by a reference shape (Sec. 4.1). Next, we add noise to the input point cloud denoted as $\mathbf{x}_0$ for $N$ time steps to generate a noisy point cloud denoted as $\mathbf{x}_N$ following Eq. 2. Then, we apply the diffusion model $D$, trained on complete data, to denoise $\mathbf{x}_N$ following Eq. 4, and obtain the final complete results $\mathbf{y}_0$ (Sec. 4.2). During the denoising process, we utilize the structure preservation guidance (Sec. 4.3.1) to preserve the original shape of the input, as well as the classifier guidance (Sec. 4.3.2) to enhance the completeness of the resulting point cloud.

Training

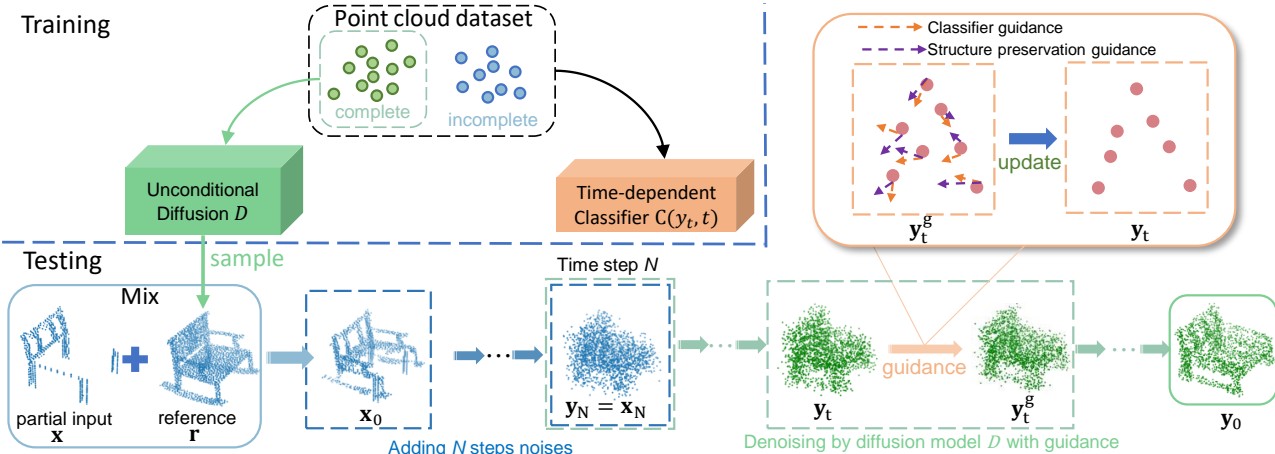

**Figure 2: Method overview. In training, we train a diffusion model on complete points, and a time-dependent binary classifier. For testing, we sample a reference shape from diffusion model, and mix it with the partial input as $\mathbf{x}_0$. We map $\mathbf{x}_0$ to the complete shape by running the forward process followed by the reverse process of the diffusion model train on complete point clouds. Structure preservation guidance and classifier guidance are proposed to facilitate the completion process.**

## 4.1 Multimodal Completion

As the aim of our method is to generate diverse completion results, we first introduce the procedure we utilize to generate multiple results. Specifically, as shown in Fig. 2, we first sample a reference point cloud $\mathbf{r}$ from the pre-trained diffusion model $D$ by denoising random gaussian noise. We then utilize a combination strategy to mix the partial input $\mathbf{x}$ with the reference shape, resulting in the input $\mathbf{x}_0$ used for instructing the generation. Our goal is to maintain the shape of the partial input while incorporating generation cues provided by the reference shape.

We denote the number of points in the complete point cloud as $n$. To reduce redundancy, we first remove repeated points from $\mathbf{x}$, and downsample it to $\frac{3}{4}n$ points if the point number is greater than $\frac{3}{4}n$, resulting in a new point cloud denoted as $\mathbf{x}'$. We then replace the $\frac{3}{4}n$ points in $\mathbf{r}$ that are nearest to $\mathbf{x}'$ with $\mathbf{x}'$ to form the mixed model input $\mathbf{x}_0$. This strategy retains most of the points in $\mathbf{x}$ while also incorporating shape information from $\mathbf{r}$. It is worth noting that the reference $\mathbf{r}$ can also be a specific shape provided by the user.

## 4.2 Progressive Completion Mapping

After obtaining the mixed input $\mathbf{x}_0$, we perform a progressive mapping from the partial input to the complete point cloud. As shown in Fig. 2, the forward process (Eq. 2) of the diffusion model perturbs $\mathbf{x}_0$ with noise. We denote the point cloud sequence derived by $N$ iterative forward steps as $\mathbf{x}_0, \mathbf{x}_1, \cdots, \mathbf{x}_N$, where $N$ is a hyper-parameter controlling the amount of noise added to the input image. Then the reverse process (Eq. 4) iteratively removes noise for $N$ steps to generate the denoised point cloud sequence $\mathbf{y}_{N-1}, \mathbf{y}_{N-2}, \cdots, \mathbf{y}_0$. Our motivation is that since the diffusion model is trained on complete point clouds, the generated point cloud $\mathbf{y}_0$ should be biased towards the distribution of complete point clouds.

While this mapping can transfer the incomplete point cloud to complete distribution, a trade-off arises when choosing the diffusion step $N$. Too little diffusion, when $N$ is small, fails to map outside

of the partial data into the complete distribution. However, too much diffusion, when $N$ is large, fails to preserve the original input shape and structure, resulting in an output that is totally different from the original partial input. Our ultimate objective is to transfer the partial point cloud to the complete one while preserving its discriminative structure.

To better achieve this objective, we introduce two guidance mechanisms operating at each time step during the denoising process to further facilitate the incomplete-to-complete mapping.

## 4.3 Guided Denoising

*4.3.1 Structure Preservation Guidance.* We introduce a structure preservation guidance which regularizes the diffusion process to better preserve the points in the partial input. Specifically, we preserve the points in the partial input by "adjusting" the points in $\mathbf{y}_t$ during denoising. We illustrate the adjustment process in Fig. 3 We first select $K$ key points in $\mathbf{x}_0$ by farthest point sampling. Key points are only selected within the points that belong to $\mathbf{x}'$ in $\mathbf{x}_0$ as described in Sec. 4.1. As the point order is fixed in the diffusion process, when we select the key points in $\mathbf{x}_0$, we could simply use the corresponding indices to find the corresponding points in $\mathbf{x}_t$ or $\mathbf{y}_t$. We denote the selected key point indices as a set $\mathcal{S}$.

We try to "push" the selected points in $\mathbf{y}_t$ to be close to those in $\mathbf{x}_t$. It means that the trajectories of the selected points will be similar in both the forward and backward pass, so that the original points can be maintained in $\mathbf{y}_0$.

We denote the $i$-th point in $\mathbf{y}_t$ as $\mathbf{y}_t^{\{i\}}$, and the $i$-th point in $\mathbf{x}_t$ as $\mathbf{x}_t^{\{i\}}$. We compute the difference between the coordinates of $\mathbf{x}_t$ and $\mathbf{y}_t$, and apply an interpolation function $f(\cdot)$ to the selected points in $\mathbf{y}_t$. The interpolation function is defined as follows:

$$f(\mathbf{y}_t^{\{i\}}) = \begin{cases} \mathbf{y}_t^{\{i\}} + \lambda(\mathbf{x}_t^{\{i\}} - \mathbf{y}_t^{\{i\}}), i \in \mathcal{S} \\ \mathbf{y}_t^{\{i\}}, i \notin \mathcal{S} \end{cases}, \qquad (6)$$

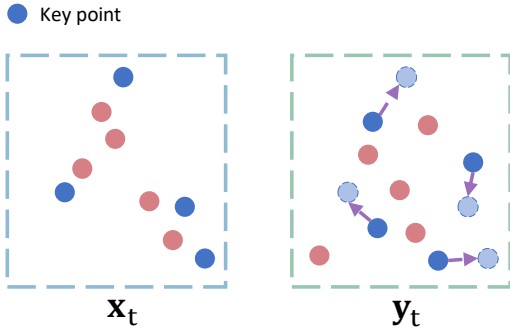

**Figure 3: Illustration of structure preservation guidance. The blue points denote the key points, and the red points denote the other points in a point cloud. Structure preservation guidance encourages the key points in $\mathbf{y}_t$ to move towards the corresponding points in $\mathbf{x}_t$.**

where $\lambda$ is a weight factor. When $\lambda$ is equal to 1, it indicates that we directly substitute the key point in $\mathbf{y}_t$ with the corresponding point in $\mathbf{x}_t$. In our experiments, we observe that higher values of $\lambda$ result in low-quality generated point cloud $\mathbf{y}_0$. This is because if the interpolation function $f(\cdot)$ alters $\mathbf{y}_t$ excessively, it may disrupt the learned reverse process of the pre-trained diffusion model.

*4.3.2 Classifier Guidance.* We introduce another guidance mechanism named classifier guidance, which aims to promote the denoised point cloud during the reverse process to be closer to the distribution of complete point clouds. Specifically, we introduce a time-dependent binary classifier $C(\mathbf{p}, t) : \mathbb{R}^{3 \times N} \times \mathbb{R} \rightarrow \mathbb{R} \in \{0, 1\}$, which predicts whether a noisy point cloud $\mathbf{p}$ is from the complete set or incomplete set. In particular, the time-dependent classifier $C(\cdot, \cdot)$ is trained on both the partial and complete point clouds with the noisy point cloud sequences $(\mathbf{x}_0, \cdots, \mathbf{x}_t, \mathbf{y}_0, \cdots, \mathbf{y}_t)$ generated from the forward diffusion process. The classification loss is denoted as $\mathcal{L}_{cls}(\mathbf{p}, t)$.

Building upon it, we regard the current point cloud $\mathbf{y}_t$ as learnable parameters, and optimize it by backpropagating the gradients according to $\mathcal{L}_{cls}$, where $\mathbf{y}_t$ is labeled as complete data. We then update $\mathbf{y}_t$ according to the gradients:

$$\mathbf{y}_t \leftarrow \mathbf{y}_t - \eta \nabla_{\mathbf{y}_t} \mathcal{L}_{cls}(\mathbf{y}_t, t), \tag{7}$$

where $\eta$ is the updating rate. In our implementation, we utilize Point-Voxel CNN [32] as the backbone of our binary classifier. The architecture details are provided in the supplementary materials.

Overall, at each reverse time step $t$, we denote the refined $\mathbf{y}_t$ as $\mathbf{y}_t^g$, which can be expressed as:

$$\mathbf{y}_t^g = f(\mathbf{y}_t - \lambda_{cls} \nabla_{\mathbf{y}_t} \eta(\mathbf{y}_t, t)), \tag{8}$$

where the classifier guidance is performed before structure preservation guidance.

Note that training the classifier with the full range of $t$ can be time-consuming. Since the range of $N$ in our experiments is limited from 0 to 100, we opt to train the classifier only using $t$ values within that range.

## 4.4 Implementation Details

We follow the existing unsupervised point cloud methods [2, 9, 58, 60, 77] and train our model separately on each category for better

quality. The number of points of the predicted complete shapes is 2048 for all datasets. For the unconditional diffusion model, we use Point-Voxel CNN [32] as the prediction backbone with the same setting in PVD [88]. The total time step of the diffusion model is $10^3$. We set the incomplete-to-complete mapping step $N$ to 25 in all experiments. We set $\lambda = 0.25$, $\eta = 0.2$, $K = 512$. Four A5000 GPUs are used for training the diffusion model. More details of the model and classifier architecture are provided in supplementary materials.

## 5 EXPEIMENTS

**Datasets.** To conduct a comprehensive evaluation, we perform experiments on both synthetic and real-world partial shapes. For synthetic datasets, we evaluate our method on 3D-EPN [14] and CRN [55] using the same training and testing splits as in ShapeInversion [77]. For 3D-EPN [14] and CRN [55] datasets, we follow the setting in MPC [60], which evaluates the multimodal completion ability on chair, plane, and table categories. For real-world datasets, we test our method on MatterPort3D [7], ScanNet [13] and KITTI [17] using the same settings as in MPC [60]. As there is no complete ground truth available in real-world datasets, we utilize the trained model on CRN [55] for testing rather than retraining our model.

**Evaluation Metrics.** We adopt the same metrics used in MPC [60] for multimodal completion quantitative evaluation. For each partial shape in the test set, we generate $k = 10$ completion results and use the following measures for quantitative evaluation: **1)** The quality of the completed shape can be evaluated using the Minimal Matching Distance (MMD), which involves computing the $L_1$ Chamfer Distance (CD) between the set of completion shapes and the set of test shapes. **2)** To measure the diversity of completion shapes for a partial input shape, we use the Total Mutual Difference (TMD). This involves summing up the differences in Chamfer distance among the $k$ completion shapes for the same partial input. **3)** Unidirectional Hausdorff Distance (UHD) is used to evaluate the fidelity of the completion to the input partial shape. This is done by calculating the average Hausdorff distance between the input partial shape and each of the $k$ completion results.

**Comparison baselines.** We present qualitative and quantitative comparisons against baseline methods. For multi-modal unpaired completion methods, we compare our method with the MPC [60] and ShapeInversion [77] both qualitatively and quantitatively. Single-modal unpaired completion methods pcl2pcl [9], C4C [58], Cai [2], P2C [11] are included for quantitative comparison reference.

## 5.1 Results on synthetic datasets

We first present the qualitative comparison of our method with other baselines on multimodal shape completion in Fig. 4 on 3DEPN and CRN datasets. We randomly sample the referenced point clouds from our unconditional diffusion model and perform our completion process. The results show that our method can generate diverse results while preserving the original shape as much as possible. Although MPC and ShapeInv can generate smooth and reasonable shapes, but their diversity is limited.

Quantitative comparison results on 3DEPN and CRN datasets are presented in Table 1. Our approach demonstrates superior performance compared to other methods, as evidenced by its highest TMD score for diversity, lowest MMD-CD score for completion

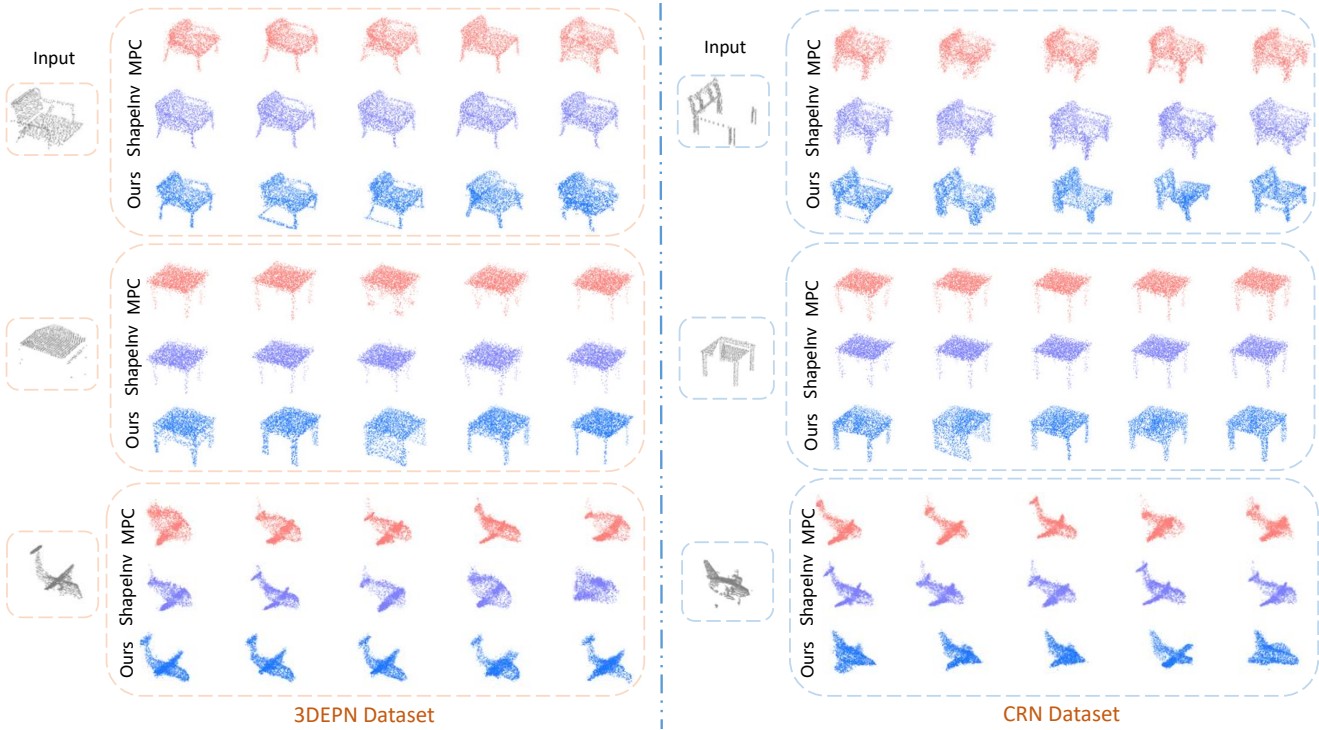

**Figure 4: Qualititive comparison of multimodal shape completion results.**

| | Methods | 3DEPN Dataset | | | | | | | | | | | | CRN Dataset | | | | | | | | | | | | |
|---|---|---|---|---|---|---|---|---|---|---|---|---|---|---|---|---|---|---|---|---|---|---|---|---|---|---|
| | | MMD-CD ↓ | | | | TMD ↑ | | | | UHD ↓ | | | | MMD-CD ↓ | | | | TMD ↑ | | | | UHD ↓ | | | |
| | | Chair | Plane | Table | Avg. | Chair | Plane | Table | Avg. | Chair | Plane | Table | Avg. | Chair | Plane | Table | Avg. | Chair | Plane | Table | Avg. | Chair | Plane | Table | Avg. |
| Single-modal | pcl2pcl [9] | 1.81 | 1.01 | 3.12 | 1.98 | 0.00 | 0.00 | 0.00 | 0.00 | 5.31 | 9.71 | 9.03 | 8.02 | 3.51 | 1.56 | 2.71 | 2.59 | 0.00 | 0.00 | 0.00 | 0.00 | 7.37 | 8.12 | 8.63 | 8.04 |
| | C4C [58] | 1.46 | 0.37 | 2.25 | 1.36 | 0.00 | 0.00 | 0.00 | 0.00 | - | - | - | - | 1.81 | 0.52 | 1.89 | 1.41 | 0.00 | 0.00 | 0.00 | 0.00 | - | - | - | - |
| | Cai [2] | 1.21 | 0.35 | 1.98 | 1.18 | 0.00 | 0.00 | 0.00 | 0.00 | - | - | - | - | 1.39 | 0.39 | 1.71 | 1.17 | 0.00 | 0.00 | 0.00 | 0.00 | - | - | - | - |
| | KT-Net [6] | 1.24 | 0.27 | 1.58 | 1.03 | 0.00 | 0.00 | 0.00 | 0.00 | - | - | - | - | 0.90 | 0.38 | 1.17 | 0.82 | 0.00 | 0.00 | 0.00 | 0.00 | - | - | - | - |
| | P2C [56] | 1.13 | 0.37 | 1.52 | 1.01 | 0.00 | 0.00 | 0.00 | 0.00 | 4.82 | 8.11 | 6.52 | 6.48 | - | - | - | - | - | - | - | - | - | - | - | - |
| Multi-modal | MPC [60] | 1.61 | 0.82 | 2.57 | 1.67 | 2.56 | 2.03 | 4.49 | 3.03 | 8.33 | 9.59 | 9.03 | 8.98 | 3.10 | 1.41 | 2.31 | 2.27 | 2.50 | **2.77** | 4.12 | 3.13 | 10.2 | 8.2 | 8.51 | 8.97 |
| | ShapeInv [77] | 1.57 | 0.85 | 2.32 | 1.58 | 2.03 | **2.11** | 4.21 | 2.78 | 7.91 | 9.26 | 8.31 | 8.49 | 2.01 | 1.32 | 1.96 | 1.76 | 2.12 | 2.27 | 4.32 | 2.90 | **8.66** | 7.67 | **8.12** | 8.15 |
| | Ours | **1.40** | **0.45** | **1.25** | **1.09** | **3.76** | 1.69 | **4.51** | **3.32** | 7.51 | 6.49 | 6.88 | 6.96 | **1.75** | **0.47** | **1.88** | **1.36** | **5.48** | 1.57 | **4.90** | **3.98** | 9.54 | **5.21** | 8.52 | **7.75** |

**Table 1: Quantitative comparison results on 3DEPN. Top two methods on each measure are bolded and underlined, respectively. MMD-CD (quality), TMD (diversity) and UHD (fidelity) presented are multiplied by $10^3$, $10^2$ and $10^2$, respectively.**

| Methods | ScanNet | | | | | | Matterport | | | | | | KITTI | |
|---|---|---|---|---|---|---|---|---|---|---|---|---|---|---|
| | TMD ↑ | | | UHD ↓ | | | TMD ↑ | | | UHD ↓ | | | TMD ↑ | UHD ↓ |
| | Chair | Table | Avg. | Chair | Table | Avg. | Chair | Table | Avg. | Chair | Table | Avg. | Car | Car |
| pcl2pcl [9] | 0.00 | 0.00 | 0.00 | 10.1 | 11.8 | 10.9 | 0.00 | 0.00 | 0.00 | 10.5 | 11.8 | 11.1 | 0.00 | 14.1 |
| MPC[60] | 1.70 | **2.40** | 2.02 | 12.1 | 10.7 | 11.4 | 1.81 | 2.81 | 2.31 | 12.1 | 10.9 | 11.5 | - | - |
| ShapeInv[77] | 1.67 | 2.13 | 1.90 | 9.3 | 11.0 | 10.1 | 2.13 | 2.99 | 2.56 | **9.50** | 10.7 | 10.1 | 3.11 | 12.5 |
| Ours | **2.51** | 1.57 | **2.04** | **8.87** | **8.21** | **8.54** | **3.17** | **3.34** | **3.25** | 10.5 | **8.28** | **9.39** | **3.34** | **11.2** |

**Table 2: Quantitative comparison results on ScanNet.**

quality, and lowest TMD score for faithfulness to the partial input. However, MPC and ShapeInv generate diverse results, but they significantly modify the original shape with a high UHD score. While Pcl2pcl can achieve good fidelity, it lacks diversity in its generated results. Furthermore, our method exhibits relatively lower diversity (TMD) when applied to the plane category, as shown in the last row of Fig. 4. This is due to our approach's ability to preserve the original shape of planes while producing plausible results. In

contrast, other methods generate diverse shapes, but many of them are implausible.

## 5.2 Results on real-scanned data.

Our pre-trained model on synthetic dataset can be directly applied on real scanned data. To evaluate the performance of our approach on real scanned data, we utilize the pre-trained model on CRN dataset. As shown in Fig. 5, our approach generates reasonable

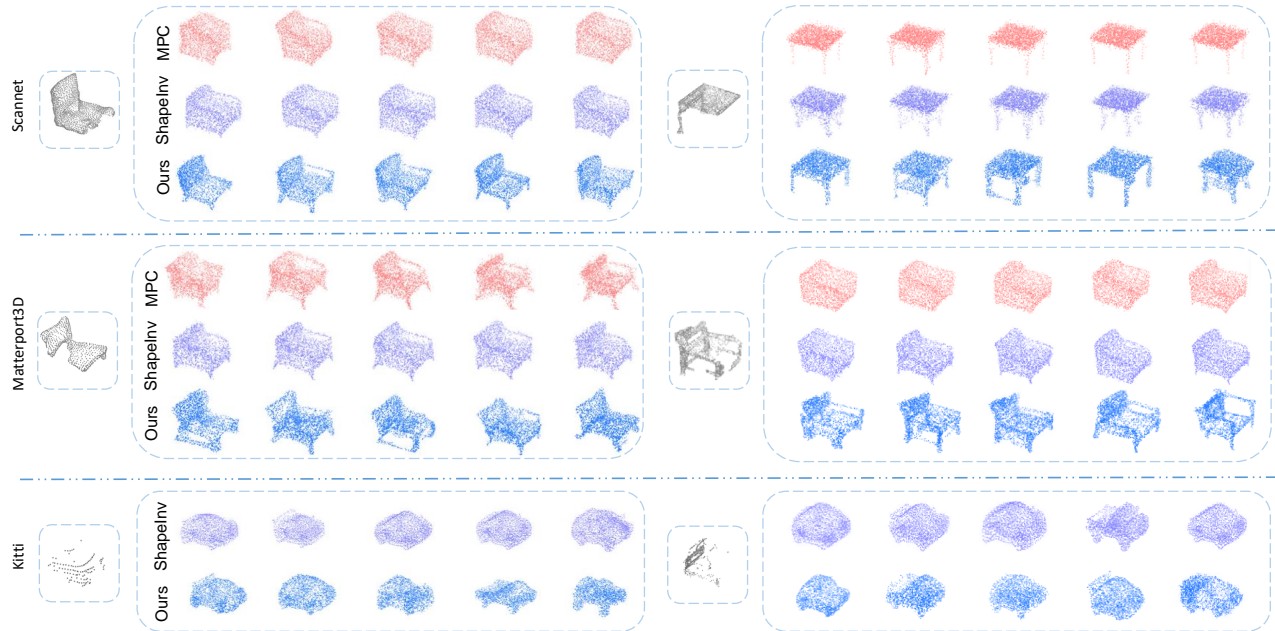

**Figure 5: Visual comparison of multimodal shape completion methods on ScanNet, Matterport3D, and KITTI datasets.**

complete shapes from the partial scans with better diversity than other methods. Table 2 presents the quantitative results on ScanNet, Matterport3D, and KITTI datasets. Our approach outperforms other baselines in terms of fidelity (lowest average UHD) and diversity (highest average TMD). There are no reported results of MPC [60] on KITTI dataset because MPC did not release a pre-trained model on car category. Also, due to the higher noise level in real-scanned data compared to synthetic data, the UHD scores are generally higher than those on synthetic datasets.

## 5.3 Referenced completion results

We show the referenced point cloud with generated completion results in Fig. 6, which enables us to complete the partial shape under a specific shape given by users. Results show that our method can generate diverse results guided by reference shape with faithful structure preservation.

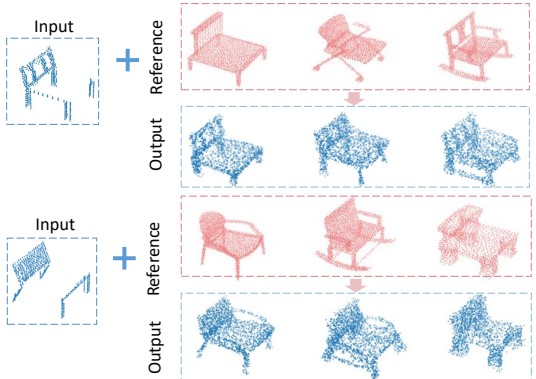

**Figure 6: Completion results guided with reference shapes.**

| Method | MMD-CD ↓ | TMD↑ | UHD↓ |
|---|---|---|---|
| w/o both guidances | 2.26 | **3.76** | 9.91 |
| w/o structure preservation | 1.72 | 3.51 | 8.70 |
| w/o classifier | 1.61 | 3.31 | 6.89 |
| with full guidance | **1.09** | 3.32 | **6.69** |

**Table 3: Effectiveness of two guidances on 3DEPN.**

## 5.4 Ablation Study

**Effectiveness of proposed two guidances.** We first provide a comprehensive evaluation of the two guidance mechanisms on 3DEPN in Tab. 3. We evaluate the impact of each guidance mechanism. We could observe that it achieves the best MMD-CD and UHD when two guidances are used. Though the results are more diverse without any guidance (i.e., w/o both guidances), they reach much higher MMD-CD and UHD indicating poor quality and fidelity.

We further conduct experiments to evaluate the proposed guidance mechanisms by varying the weight factor $\lambda$ for the structure preservation guidance and the updating rate $\eta$ for the classifier guidance. Results are presented in Fig. 7 and 8. During the evaluation, we fixed the sampled reference shapes.

Fig. 7 shows the evaluation results for structure preservation guidance. We observe that the results become more faithful to the partial input (lower UHD) when $\lambda$ become larger, but the diversity is decreased (lower TMD). The completion quality (MMD-CD) does not show obvious changes when $\lambda$ is smaller than 0.5, but it significantly decreases when $\lambda$ becomes larger, indicating that it will disrupt the reverse trajectory when we excessively alter $\mathbf{y}_t$. If we do not involve structure preservation guidance ($\lambda$=0), it could still generate reasonable results but with less fidelity.

In Fig. 8, we present the evaluation results for classifier guidance, along with the computed average classification loss $\mathcal{L}_{cls}$ of the final results after applying classifier guidance. The results indicate that

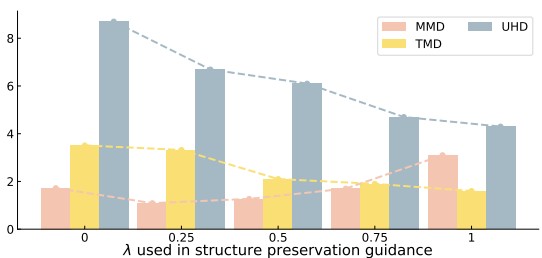

Figure 7: Evaluation of the proposed structure preservation guidance, where the x-axis is the hyper-parameter $\lambda$.

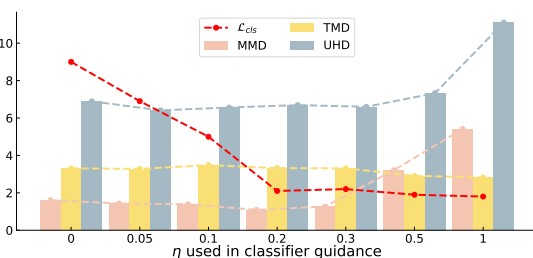

Figure 8: Evaluation of the proposed classifier guidance, where the x-axis is the hyper-parameter $\eta$.

setting the value of $\eta$ from 0.1 to 0.3 yields lower MMD-CD scores with better quality, and classifier guidance has a minimal impact on the diversity and fidelity of the generated shapes. The results deteriorate significantly when $\eta$ exceeds 0.5, while $\mathcal{L}_{cls}$ remains at a low level. We believe that this is due to the dominance of classifier guidance in the generation process, which obstructs the denoising effect of the diffusion model.

Besides, we visualize the binary classifier output distribution in Fig. 9 with a chair example. Interestingly, we could observe that as the noises are added to the incomplete chair, the classifier tends to classify it as complete. This is because noises are randomly added which may have filled the missing parts to some extent. During the denoising process using the classifier guidance, the classifier could finally regard the denoised sample as complete.

**Evaluation of the diffusion step $N$.** An important consideration for our method is selecting an appropriate value for $N$ that balances completeness and fidelity. We evaluate the impact of varying $N$ on our results, as shown in Fig. 10 (a). We can observe that the generation quality cannot be guaranteed when $N$ is set to 5 or 10, even though the UHD is low. This is because the results are still very similar to the original input. Both the quality and fidelity improve when we set $N$ to 25. However, when $N$ becomes larger, particularly larger than 40, it generates shapes with good diversity, but both MMD-CD and UHD become high, indicating that the generated shapes are significantly different from the input. As our implementation follows that of DDPM [22], the results often converge to noise in the early time steps. Therefore, we set $N$ to a small value (e.g., 25).

**Impacts of the value of $K$.** We investigate the impact of the value of $K$ used in the structure preservation guidance. As the total number of points in $\mathbf{x}'$ is $\frac{3}{4}n$, which equals 1532 (0.75 x 2048), we set

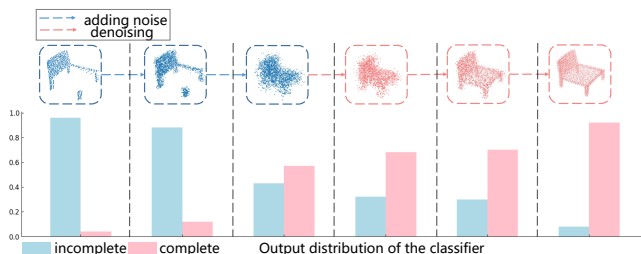

Figure 9: Visulization of the classifier output during both the forward and backward process of the diffusion model.

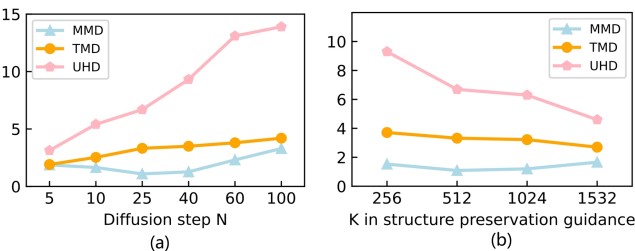

Figure 10: Evaulation on 3DEPN dataset with different $N$ and $K$ in proposed two guidances.

$K$ to [256, 512, 1024, 1532] to evaluate the performance on 3DEPN. The value of $\lambda$ is set to the default value of 0.2. The quantitative results are presented in Fig. 10 (b). We can see that high UHD scores are achieved when $K$ is set to 256, indicating that less structure is preserved. However, when $K$ exceeds 1024 (half of the total point number), it results in low completion quality with less diversity. This suggests that selecting too many points for original structure preservation can disrupt the transformation process of the diffusion.

## 6 CONCLUSION

In this paper, we introduce a novel unpaired multimodal shape completion approach. By employing an unconditional diffusion model that was trained on a complete point cloud dataset, we execute the forward and reverse processes to map the partial point cloud test data to the complete point cloud. Unlike previous methods that use a latent feature space to transform partial shape encoding into a complete one, our approach ensures a smoother completion transformation in the coordinate space. Furthermore, we have improved the diffusion model by incorporating two guidance mechanisms that assist in transferring the partial point cloud to the complete one and maintaining its original structure. Experimental results show our method produces shapes that maintain their original structure while also exhibiting better diversity compared to other methods.

## 7 ACKNOWLEDGEMENT

This research was supported by the National Natural Science Foundation of China (NSFC) under Grant U22A2094. This research was also supported by the advanced computing resources provided by the Supercomputing Center of the USTC.

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
