# OpenReview forum: "Reverse2Complete: Unpaired Multimodal Point Cloud Completion via Guided Diffusion"
_acmmm.org/ACMMM/2024/Conference — MM2024 Poster_

### Official Review · Reviewer_nrpm · 2024-04-27
**This paper proposes a point cloud completion method based on a diffusion model. Initially, a fixed number of points are sampled from the point cloud, and then the diffusion model is employed for denoising and completion.**

**Rating:** 4
**Confidence:** 1

**Review:**

This paper introduces a new point cloud completion method based on a diffusion model, which utilizes structure preservation guidance and classifier guidance to aid in generation. In fact, I believe this is a conditional generation method.

Strengths:
1. The strength of this paper lies in the introduction of partial input points for structure preservation guidance, which seems effective and potentially novel for this task. However, the guidance based on classifier has been widely discussed for two years and is difficult to consider as an advantage.

Weakness:
1. Diffusion-based models tend to introduce more sampling, which should result in greater computational complexity compared to existing methods, but the performance advantages over existing methods are not obvious. The authors also did not explicitly compare model complexity and inference time.
2. The authors lack discussion on point cloud generation works based on diffusion models, which are closely related to this paper.
3. Some of the images in this paper seem to be of low quality.

Since I am not very familiar with this task, I will revise my rating after referring to the comments of other reviewers.

**Summary:**

This paper introduces a new point cloud completion method based on a diffusion model, which utilizes structure preservation guidance and classifier guidance to aid in generation. In fact, I believe this is a conditional generation method. Since I am not very familiar with this task, I will revise my rating after referring to the comments of other reviewers.

**Strengths:**

1. The strength of this paper lies in the introduction of partial input points for structure preservation guidance, which seems effective and potentially novel for this task. However, the guidance based on classifier has been widely discussed for two years and is difficult to consider as an advantage.

**Limitations:**

1. Diffusion-based models tend to introduce more sampling, which should result in greater computational complexity compared to existing methods, but the performance advantages over existing methods are not obvious. The authors also did not explicitly compare model complexity and inference time.
2. The authors lack discussion on point cloud generation works based on diffusion models, which are closely related to this paper.
3. Some of the images in this paper seem to be of low quality.

**Suitability:**

2

---

### Official Review · Reviewer_FCfK · 2024-05-20

**Rating:** 4
**Confidence:** 4

**Summary:**

This paper proposes a novel method for unpaired shape completion. To further improve the method, the paper designs two mechanisms to facilitate mapping the partial point cloud to the complete one while preserving its original structure. The author has conducted a lot of experiments to demonstrate the method's effectiveness.

**Strengths:**

1. The paper is well-organized and clearly written, making it easy to understand.
2. This paper proposes a novel unpaired point cloud completion method, eliminating the need for one-to-one data during training.
3. The paper introduces two new mechanisms to enhance point cloud completion performance, and experiments have validated their effectiveness.

**Limitations:**

1. There are some minor issues with the format, such as Figure 7.
2. This paper does not test on the PartNet dataset.
3. No comparison with PVD was conducted.

Further comments:
1. Could you please provide a more detailed explanation of why Classifier Guidance is effective?
2. Could you please provide a detailed explanation of the experimental process for the one-to-one completion of the PCN dataset?
3. Could you compare it with implicit methods such as DiffComplete[1]?

Reference:
[1] Ruihang Chu, Enze Xie, Shentong Mo, Zhenguo Li, Matthias Nießner, Chi-Wing Fu, Jiaya Jia: DiffComplete: Diffusion-based Generative 3D Shape Completion. NeurIPS 2023

**Suitability:**

3

---

### Official Review · Reviewer_2tgV · 2024-05-21

**Rating:** 2
**Confidence:** 3

**Summary:**

This work points out the limitations of existing methods, which lack diversity in the generated shapes. To address this problem, they propose an unpaired multimodal shape point cloud completion approach that directly operates on point coordinate space. The proposed method takes advantage of the pre-trained diffusion model to aid the generative process. Additionally, this work augments the diffusion model by introducing two guidance mechanisms, namely Classifier and Structure Preservation Guidance, to facilitate mapping the partial point cloud to the complete one while preserving its original structure.

**Strengths:**

1. This work conducts extensive experiments to explore hyperparameters, such as Fig.7-Fig.10

2. The motivation of this work is clear: to use unpaired data and attempt to generate diverse results.

**Limitations:**

1. What is y_t? Is it the same as y_N?

2. Why combine the incomplete point cloud x with the reference point cloud r? Since r is randomly generated, r and x might be completely different. In Figure 6, all inputs and references are chairs. Can the pretrained diffusion model only generate chairs, or is it capable of producing other categories as well?

3. Why replace 3/4 of the points in r? Illustrating this process and discussing the motivation behind it is essential.

4. In the completed results, the parts that were not missing are supposed to be identical to the original input. Diversity should be limited to the missing parts. However, the shape of some results generated by the proposed method is entirely different from the original input, which does not make sense. Refer to Figures 2 and 4 for examples.

**Suitability:**

2

---

### Official Review · Reviewer_NW7Z · 2024-05-25

**Rating:** 5
**Confidence:** 3

**Summary:**

The paper proposes a diffusion-based framework for unsupervised point cloud completion. The proposed method executes the forward and reverse processes to map the partial point cloud test data to the complete point cloud. Two types of guidance are introduced to preserve the input points.

**Strengths:**

+ The way to use diffusion models is novel and interesting. It provides an elegant way to do unsupervised completion with pre-trained diffusion models. Since the diffusion model can be trained on large-scale 3D datasets, the proposed method shows potential to complete in-the-wild point clouds. This makes the paper a good trial towards zero-shot point cloud completion.
+ The proposed method shows state-of-the-art multi-modal completion performance.
+ The proposed method also supports completion with a reference shape provided by users.
+ This paper is well-written.

**Limitations:**

- It seems like the proposed method only supports single-category training. If the diffusion model is trained on multiple categories, the generated reference shape can not be well controlled.
- The classifier guidance paper should be cited.

**Suitability:**

2

---

### Meta-Review · Area_Chair_QPEG · 2024-06-28

**Recommendation:** Accept (Poster)
**Confidence:** 4

**Metareview:**

This paper receives mixed reviews of 3 postive and 1 negative ratings. After carefully reading both the reviews and the paper, the AC agrees with the paper's novelties in using diffusion models for point cloud completion task. Moreover, the authors faithfully prepared the rebuttal, which successfully addressed most of reviewers' concerns. In general, the AC leans towards an acceptance, but would like to encourage the authors to further improve the quality of camera ready paper, e.g., providing more discussions on concurrent diffusion-based completion works, presenting better rendered figure visualizations, etc.

---

### Meta-Review · Senior_Area_Chairs · 2024-07-10

**Recommendation:** Accept (Poster)
**Confidence:** 4

**Metareview:**

This paper received mixed ratings initially. After rebuttal, three reviewers tend to accpt the paper while one reviewer increased the score from WR to BR. SAC and AC carefully checked the reviews and rebuttal and recommend acceptance of the paper.